# CALMS: Modelling the long-term health and economic impact of Covid-19 using agent-based simulation

Kate Mintram[1]◉*, Anastasia Anagnostou◉[1]◉*, Nana Anokye[2], Edward Okine[2], Derek Groen[1], Arindam Saha[1], Nura Abubakar◉[1], Tasin Islam[1], Habiba Daroge[1], Maziar Ghorbani[3], Yani Xue[1], Simon J. E. Taylor[1]

**1** Modelling and Simulation Group, Department of Computer Science, Brunel University London, Uxbridge, London, United Kingdom, **2** Global Public Health, Department of Health Sciences, Brunel University London, Uxbridge, London, United Kingdom, **3** Department of Electronic and Computer Engineering, Brunel University London, Uxbridge, London, United Kingdom

◉ These authors contributed equally to this work.

* kate.mintram@brunel.ac.uk (KM); anastasia.anagnostou@brunel.ac.uk (AA)

**Data Availability Statement:** All data files are available from https://doi.org/10.17633/rd.brunel.19350518 The software is available on https://gitlab.com/anabrunel/calms.

## Abstract

We present our agent-based CoronAvirus Lifelong Modelling and Simulation (CALMS) model that aspires to predict the lifelong impacts of Covid-19 on the health and economy of a population. CALMS considers individual characteristics as well as comorbidities in calculating the risk of infection and severe disease. We conduct two sets of experiments aiming at demonstrating the validity and capabilities of CALMS. We run simulations retrospectively and validate the model outputs against hospitalisations, ICU admissions and fatalities in a UK population for the period between March and September 2020. We then run simulations for the lifetime of the cohort applying a variety of targeted intervention strategies and compare their effectiveness against the baseline scenario where no intervention is applied. Four scenarios are simulated with targeted vaccination programmes and periodic lockdowns. Vaccinations are targeted first at individuals based on their age and second at vulnerable individuals based on their health status. Periodic lockdowns, triggered by hospitalisations, are tested with and without vaccination programme in place. Our results demonstrate that periodic lockdowns achieve reductions in hospitalisations, ICU admissions and fatalities of 6-8% compared to the baseline scenario, with an associated intervention cost of £173 million per 1,000 people and targeted vaccination programmes achieve reductions in hospitalisations, ICU admissions and fatalities of 89-90%, compared to the baseline scenario, with an associated intervention cost of £51,924 per 1,000 people. We conclude that periodic lockdowns alone are ineffective at reducing health-related outputs over the long-term and that vaccination programmes which target only the clinically vulnerable are sufficient in providing healthcare protection for the population as a whole.

**Funding:** KM, AA, NA, EO, DG, AS, NA, TI, HD, MG, YX and ST have received funding from the EU Horizon 2020 STAMINA project No. 883441 (https://cordis.europa.eu/project/id/883441).

**Competing interests:** The authors have declared that no competing interests exist.

## Introduction

Infectious diseases have the potential to result in serious cross-border public health threats. Global pandemics can emerge at any time as demonstrated by the influenza (H1N1) outbreak in 2009, the 2011 Escherichia Coli outbreak in Germany, the Ebola virus in 2014, Zika virus in 2016 and West Nile virus in Southern and Eastern European countries in 2019 [1]. Most recently in December 2019, Covid-19 emerged in Wuhan, China and has since spread to almost every country in the world resulting in a global health and economic crisis. Reliable predictive models have been used to understand the evolving landscape of Covid-19 throughout the pandemic and assess the impacts of different interventions. Governments have been using results from these models to inform their decision making.

There has been a surge in modelling studies attempting to predict the spread of Covid-19. The most notable models used to inform the UK government at the start of the pandemic included mathematical and agent-based models developed by Imperial College London [2], and transmission models developed by PHE/Cambridge [3]. A large number of models have since been developed which primarily aim to assess the spread of Covid-19 under varying social distancing measures [4–6]. However, these models generally do not consider the effects of individual health status (comorbidities and other risk factors) on disease outcome, and since many models focus on assessing the spread of the virus, most do not assess the long-term or lifelong health effects of Covid-19 on populations. As well as the health effects, it is also critical to understand the economic effects of both the pandemic itself and the proposed government interventions. It was estimated that by the end of September 2020, the pandemic had cost the UK in the region of £317.4 bn [7], and there is still a need to understand the longer-term economic impacts of the virus.

Whilst the acute effects of the pandemic are the most important consideration when countries are in a state of emergency to reduce immediate hospitalisations and deaths, it is vital to consider the chronic long-term health and economic effects of government interventions as the pandemic progresses. The direct long-term effects of Covid-19 infections on individual health, healthcare services, and healthcare costs have so far been largely overlooked as the focus has been on quantifying the acute short-term effects of the viral spread, for example to inform on hospital and ICU capacity. In addition, Long Covid, a long-lasting disorder that arises following infection with SARS-CoV-2 [8], is estimated to be affecting two million people in the UK [9]. The number of people developing Long Covid over time should therefore be an important consideration when assessing individual health and associated healthcare costs.

Disease epidemics can be modelled using a variety of modelling approaches, including mathematical models, discrete-event simulation modelling, machine learning techniques, and agent-based modelling. The benefits and shortfalls of each of these modelling approaches are discussed in detail in Mahmood et al. [10]. Currie et al. [11] identified the decisions that these modelling approaches have been supporting at the early stages of the coronavirus pandemic. In particular, agent-based models have become increasingly popular for simulating the emergence of infectious diseases. They incorporate interactions between individuals (agent-agent) and their environment (agent-environment), and these individual-level interactions and behaviours result in the emergence of complex phenomena [12]. In addition, individuals can generate their own life-histories, and these life histories can be used to determine epidemiological risk and disease outcomes. Agent-based models also allow the direct effects of government interventions (pharmaceutical and non-pharmaceutical) on individuals to be simulated whilst considering uptake rates and public adherence.

We present our Coronavirus Lifelong Modelling and Simulation (CALMS) agent-based model which simulates the lifelong health and economic impacts of Covid-19 on a population,

considering individual comorbidities when establishing disease outcomes. This feature allows for studying the effects of interventions to targeted population subgroups. Individuals in the model develop Covid-19 according to a dynamic infection probability which is dependent on the number of infected agents in the population, as well as the transmission probability of the virus. QCovid® is an evidence-based risk algorithm that uses a range of factors such as age, sex, ethnicity and existing medical conditions to predict risk of hospitalisation and death from COVID-19, and this calculator is embedded into the model to predict the disease severity of infected agents (https://qcovid.org/) [13]. QCovid has shown high levels of validity in English and Scottish cohorts [14, 15] and is a 'living' risk prediction model which is being updated as the understanding of COVID-19 increases.

CALMS is implemented in Repast Simphony (https://repast.github.io/) [16], a free and open source agent-based modelling toolkit, and written in Java. Repast Simphony provides a graphical user interface, where the user can change experimental parameters (e.g., population characteristics and Covid-19 interventions) and monitor the progression of the simulation. The model can also run as a batch application that can be integrated in a distributed computing infrastructure. The model was designed to be adaptable for future pandemics, including variants of Covid-19.

The aim of this paper is to demonstrate our approach and present the long-term health and economic impact of Covid-19 in the UK. To the best of our knowledge, this is the first study that considers demographic, health, socio-economic and cost data and predicts lifelong risks of non-communicable diseases and the impact of Covid-19 and Long Covid based on comorbidities for the lifetime of a cohort.

## Materials and methods

### Model overview

The Coronavirus Lifelong Modelling and Simulation (CALMS) model is an agent-based model that predicts the long-term health and economic impact of Covid-19, and associated preventative and/or therapeutic interventions, on a population or a subgroup of a population. Three highly prevalent non-communicable diseases (NCD) are modelled using established risk estimation algorithms. Namely, Coronary Heart Disease (CHD), stroke and type 2 diabetes (T2D), all of which are comorbidities for Covid-19 [13]. The risk of developing CHD and stroke are estimated as a probability of cardiovascular cisease (CVD) incidents. The risk for CVD is calculated using QRisk [17], while the risk of developing diabetes is determined using QDiabetes [18]. Incidences of these three conditions are functions of each individual's characteristics, including their physical activity status. CALMS is an extension of the Physical Activity Lifelong and Simulation (PALMS) [19] model that generates the lifelong physical activity trajectory of individuals and adjusts their risk for CVD and T2D accordingly. We therefore utilise this feature to generate a more comprehensive picture of our population. As well as these comorbidities, the model generates a life-history for each individual in the population at initialisation which includes age, sex, systolic blood pressure, type 1 and type 2 diabetes, and kidney disease. These life-history characteristics, along with the development of CHD, stroke, and T2D, determine the severity of Covid-19 infection according to the QCovid risk estimation algorithm (https://qcovid.org/) [13].

CALMS also estimates the impact of delivering government interventions over a period of time. It considers uptake levels and tracks the characteristics of individuals, along with the healthcare costs associated with Covid-19 infection and the economic costs of implementing interventions.

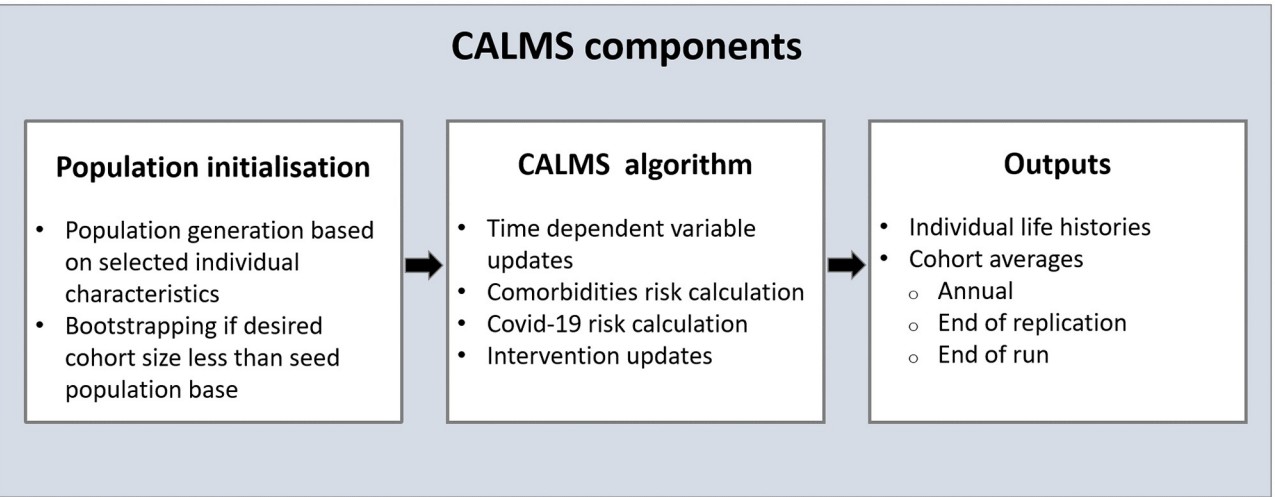

**Fig 1. CALMS high level components.**

## Model description

CALMS consists of three main components: the *population initialisation* component, the *CALMS algorithm* component and the *Outputs* component (see Fig 1). At *population initialisation*, the population agents of the simulation are generated. Individual characteristics can be selected in order to generate the population group of interest. Bootstrap sampling is applied, if the desired cohort size is less than the seed population base. The *CALMS algorithm* performs various calculations per individual at each simulation time step. It updates the time dependent variables and the risks and costs related to comorbidities, Covid-19 and any active interventions. CALMS records outputs at individual and cohort levels.

**Population initialisation.** The model includes a sample dataset of individuals drawn from health surveys. Each record represents an individual in the population and its characteristics. Population characteristics include demographics as well as socio-economic and health characteristics such as age, sex, ethnicity, education, deprivation, smoking history, physical activity status, body mass index (BMI), systolic blood pressure, hypertension treatment, high-density lipoprotein (HDL) ratio, type 1 and type 2 diabetes (T1D and T2D), cardiovascular disease (CVD), rheumatoid arthritis, chronic kidney disease and depression as well as family history of T2D and CVD. These characteristics reflect the potential determinants and commorbities of Covid-19. Currently, the CALMS population dataset includes 9,594 records that reflect the English population over five years of age based on the 2012 Health Survey for England [20]. A cohort comprised of population subgroups can be generated. The selection is based on the individual characteristics i.e., by age, sex, medical history, etc. and can include one or a combination of these characteristics. Subgroups can be selected both as a population base and as a target population group for an intervention. The intervention target population subgroup can be the whole population base or a subset of it. The population cohort size can also be selected at initialisation. If the selected cohort size is greater than the population dataset, a synthetic population is generated using sampling with replacement to inflate the cohort size based on the seed population.

Table 1 shows a summary of the sample population dataset. Population data is secondary, fully anonymised and publicly available from the UK Data Service [21].

**Table 1. Selected baseline characteristics of the initial population.**

| Seed population summary statistics (n = 9,594) | |
|---|---|
| Female, n (%) | 5,268 (54.9) |
| Age, (years) median (25th, 75th percentile) | 45 (27, 63) |
| Ethnicity, n (%) | |
| • White | 8,470 (88.3) |
| • Black | 247 (2.5) |
| • Asian | 642 (6.6) |
| • Other | 235 (2.4) |
| BMI, median (25th, 75th percentile) | 25.95 (22.21, 29.72) |
| CVD history, n (%) | 1,033 (10.8) |
| Type 2 diabetes, n (%) | 527 (5.5) |

**CALMS algorithm.** Here we describe the CALMS algorithm and the details for each sub-model. The diagrammatic representation of the algorithm is shown in Fig 2 For each alive individual in the simulation and at every simulation time step (except where specified), certain calculations are implemented as explained below. The simulation time step is one day. Input parameters, their values and sources where these values are derived from are shown in Table 2. All input parameters are set in a CSV file which is read at initialisation.

• *Calculate risks of developing NCDs*

CALMS considers risks for developing NCDs throughout the lifetime of individual. The risks of developing CVD and T2D are calculated according to the established QRisk2 [17] and QDiabetes [18] algorithms, respectively. The risk of coronary heart disease (CHD) or a stroke episode is a probability of a CVD event. CVD and T2D are considered as comorbidities of Covid-19. This sub-model additionally calculates the risks of developing depression and MSI, as described in Anagnostou et al. [19]. Both of these NCDs contribute to an agent's death probability but are not considered comorbidities of Covid-19. The NCD risks are calculated every three months. These risks are then adjusted considering the physical activity status of the individual. All risks are a function of the agent's demographic, socio-economic and medical characteristics.

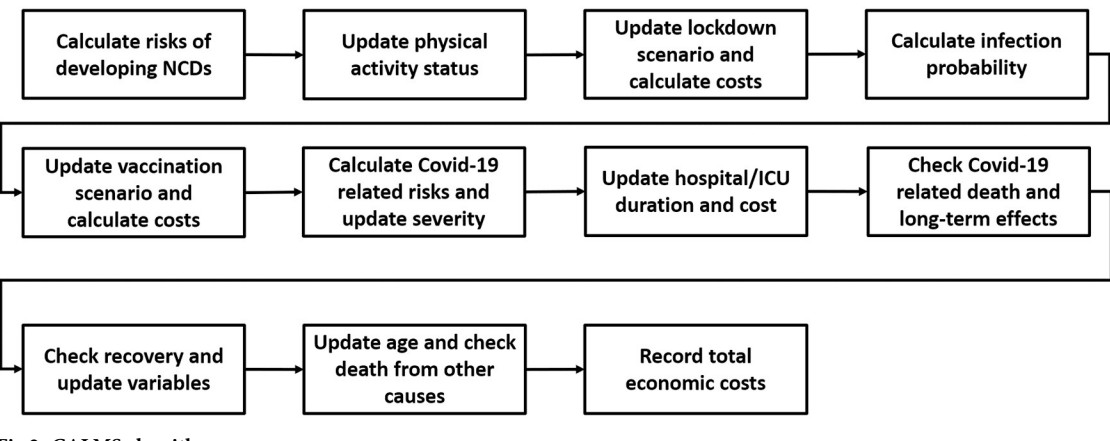

**Fig 2. CALMS algorithm.**

**Table 2. CALMS input parameter names and values, sources and additional information.**

| Parameter | Value | Additional information | Reference |
|---|---|---|---|
| Transmission Probability | 0.022 | Calculated using an R0 value of 2.2–2.7 and N contacts 10.8. | [22, 23] |
| N Contacts (no lockdown) | 10.8 | | [24] |
| N Contacts (lockdown) | 2.8 | | [24] |
| Exposure Period (days—mode (min, max)) | 3.5 (0,12) | Calculated in the model using a triangular distribution. | [25] |
| Infectious Period (days) | 9 | Considers a pre-symptomatic infectious period of 2 days. | [26, 27] |
| Time from infectious to hospitalisation (days) | 9 | Considers a pre-symptomatic infectious period of 2 days. | [28] |
| Hospital/ ICU LOS (days—mean± sd) | 12.8 (13.4) / 13.2 (13.4) | | [29] |
| ICU Probability | 0.17 | | [30] |
| Death Probability (ICU) | 0.32 | | [30] |
| Death Probability (Hospital) | 0.26 | | [30] |
| Long Covid Probability | 0.033 | Clinical data based on reported infections. Parameter value considers that the CDC estimate number of actual cases to be 4.3 times reported cases. | [31, 32] |
| Time to reinfection (months) | 6–12 | Data suggests possible reinfection after 6 months, modelled as a range between 6 and 12 months for stochasticity. | [33] |
| Duration of long covid (weeks) | 4–77 | The maximum duration of Long Covid is currently unclear, data suggests at least a year. | [34] |
| Vaccine immunity | 0.93 | | [35–37] |
| Vaccine Uptake | 0.96 | | [38] |
| Hospital/ ICU cost (£) | 797/£)1681 | | [39] |
| Vaccine cost per capita (£) | 1.57 | Cost only includes actual cost of the vaccine dose (no R&D, administration etc). | [40] |
| Lockdown cost per capita (£) | 17.53 | Calculated using the fall in GDP between Apr–June 2020 during lockdown restrictions (19.5%). | [41] |
| QCovid multiplication factor | 9 | | (*) |

[*] The multiplacation factor has been derived from calibrating the model so we calculate only the risk for hospitalisation using the QCovid risk algorithm. The method is explained in the experimental design section.

- *Update physical activity status*

  Physical activity as a key risk moderating factor for both CVD and T2D and consequently for Covid-19. Hence these risks are adapted in CALMS to incorporate the relative risk for developing a condition based on the physical activity status of individuals. Each agent in the model has a baseline physical activity status. Physical activity levels are classified into three categories using minutes of vigorous physical activity (MVPA) per week as a measure: Inactive (0 < physical activity status < 85 min per week); Moderately active (86 < physical activity status < 425 min per week); and Very active (physical activity status > 426). Individuals can switch activity categories during their lifetime. The formulae for calculating physical activity changes over time are shown in Anagnostou et al. [19]. The trajectory of lifelong physical activity is a function of previous activity levels as well as the characteristics of the agent.

- *Update lockdown scenario and calculate costs*

  A lockdown can be introduced if the global conditions (start date, end date) and agent characteristics (age, gender) meet the criteria set at initialisation. A lockdown carries a specific cost per capita per individual undertaking a lockdown and aims to reduce the number of exposures an agent encounters, considering their likelihood of adhering to the lockdown, which subsequently reduces their infection probability. The number of contacts individuals come into

contact with during a lockdown was quantified from the CoMix survey undertaken by Jarvis et al. [24].

- *Calculate infection probability*

All agents are susceptible to Covid-19 infection at initialisation. An agent's infection probability is determined by the probability of infection per exposure (i.e., transmission probability) and the number of exposures per time step according to the equation Eq (1):

$$I_p = T_p C_n C_I \qquad (1)$$

Where $I_p$ denotes infection probability, $T_p$ denotes the transmission probability of the virus, $C_n$ denotes the number of contacts each agent comes into contact with each day, and $C_I$ denotes the proportion of agents in the cohort which are infectious. The transmission probability $T_p$ is constant for a specific virus variant. The infection probability depends on the infectious population and the number of contacts an individual has per time unit (day), as per Eq 1.

If the infection probability is greater than a random number drawn between 0 and 1, an agent becomes infected and is no longer susceptible. The exposure (i.e., pre-infectious) period of the virus for each infected agent is then calculated from a triangular distribution. If an agent is not infected, the following disease related steps are not implemented and they will proceed to updating their age, checking death from other causes and recording their total costs.

An agent becomes susceptible to Covid-19 infection again 6–12 months after the previous infection [33].

- *Update vaccination scenario and calculate costs*

A vaccination programme can be introduced if the global conditions (start date, end date) and agent characteristics (age, gender) meet the criteria set at initialisation. A vaccine carries a specific economic cost and aims to reduce the risk of agents developing severe or critical symptoms. Agents decide if they are going to uptake the vaccine, according to a given uptake probability, and their risk is adjusted accordingly. The proportion of the cohort that are vaccinated per day is specified as input parameter and determines the speed of the vaccination programme.

Vaccinated agents acquire full immunity following a given time period to account for the time between the two vaccine doses as well as the time lag between the administration of the second dose and obtaining full immunity. Vaccination immunity is assumed to last the same amount of time as natural immunity. Booster vaccinations are administered every 6 months, however this time period can be set as an input parameter.

- *Calculate risks and update severity*

The risk of developing mild or severe symptoms is calculated given an agent's characteristics (e.g. age, sex, ethnicity, bmi) and comorbidities (e.g., spb, CHD, T2D, T1D, stroke, kidney disease) according to the QCovid risk estimation algorithm [13]. If an agent has severe symptoms, their risk of developing critical symptoms is calculated according to a given probability of being admitted to ICU [30]. Vaccine immunity alters this risk probability accordingly (set to 0 if there is no vaccination programme in place or the conditions for vaccine uptake are not met). Agents infected with mild coronavirus have a set infectious period of 9 days; severe or critical cases are infectious for the duration of their time in hospital/ICU.

- *Update hospital/ICU duration and cost*

   Agents with severe or critical disease are admitted to hospital or ICU, respectively, following a set time period between exposure to the virus and hospitalisation. The length of stay is calculated from a gamma distribution and an associated cost of stay is calculated.

- *Check Covid-19 related death and long-term effects*

   Agents in hospital or ICU die according to a given daily death probability. Agents with mild, severe or critical disease can develop long-term health effects (Long Covid) according to a given probability. All infected agents have the same probability of developing Long Covid, and Long Covid does not currently affect risks for developing T2D or CVD.

- *Check recovery and update variables*

   If an agent is discharged from hospital or ICU or displayed mild symptoms and is no longer infectious, they are recovered and all disease related variables are reset.

- *Update age and check death from other causes*

   All living agents in the simulation age and check if death has occurred from non Covid-19 related causes, including depression, musculoskeletal injuries (MSI), CHD, diabetes and stroke. Baseline mortality according to life tables for the age and sex groups is also considered. Costs of NCD events are additionally calculated as described in Anagnostou et al. [19]. This step occurs every three months.

- *Record total economic costs*

   Total healthcare costs (hospital and ICU) and intervention costs (vaccination and lockdown) for the cohort are summarised.

   **Outputs.** CALMS records outputs at individual and cohort levels. At the end of each simulation replication, individual life histories for all agents are generated, including all events that occurred on a specific agent in their lifetime. This feature requires to keep a memory of all events per individual and therefore is computationally expensive. For this reason, it is optional and can be selected only if the objectives of the simulation study require it. CALMS also generates a text sink which records daily outputs of Covid-19 related global parameters.

   At cohort level, for each simulation replication annual and end of simulation averages are recorded. The annual outputs can be used to calculate discounted costs and QALYS and therefore applied to further cost-benefit analysis. At the end of the simulation run, cohort averages for all replications are recorded.

   Each simulation replication run ends either when the selected runtime is reached or when the whole cohort has died. By default, all population averages (annual, end of replication and end of run) are exported to CSV files.

## Experimental design

### Motivation

To demonstrate the validity and capabilities of CALMS, we conducted two sets of experiments. The purpose of which was to validate CALMS and to study the long-term effects of Covid-19 on a UK population, respectively.

   For model validation, we ran simulations retrospectively and compared our results with known Covid-19 outcomes. We ran 100 replications on a cohort of 10,000 agents for a period of 243 days from 31st January 2020 to 30th September 2000. Outputs of total number of infections, hospitalisations, ICU admissions, fatalities and Long Covid cases were recorded each

day of the simulation to match the field data. We also validated our results against risk factors for Covid-19 fatalities and ICU admissions. Outputs of percentage of deaths by risk factor and percentage of ICU admissions by BMI were cumulative over the simulated time period.

For the case study, we ran 100 replications on a cohort of 1,000 agents for a period of 80 years. Annual outputs of each endpoint were used for analysis. For all simulations, Covid-19 was introduced into the population at initialisation, with 10 individuals being initially infected.

## Model validation

**Hospital/ICU admissions and death rates.**   Model validation was undertaken using official government data for England [42]. Field data for hospitalisations, ICU admissions and fatalities between 20th March 2020 and 30th September 2020 were compared to model outputs starting on 2nd April 2020. The model is currently developed for the wild-type virus and does not consider variants at this point. Therefore, validation was undertaken up to 30th September 2020 as the alpha variant began to affect infection rates after this time [43]. The model was initialised to represent the 31st January 2020, when the first cases of Covid-19 were detected in the UK. The national lockdown was modelled from 24th March 2020 using data from the CoMix survey which measured contact patterns weekly for different age groups from March 2020 to March 2021 to estimate the impact of the different lockdown policies in the UK [44]. The QCovid algorithm calculates an individual's risk of contracting and becoming hospitalised from Covid-19. Since the model already calculates the risk of developing Covid-19, we calibrated the model using a multiplication factor which allows the QCovid algorithm to calculate only the risk of being hospitalised. The government data for hospitalisations used to validate the model was split such that the data between 20th March 2020 and 20th April 2020 was used for calibration. A parameter sweep was undertaken initially by using multiplication factors of 0, 5, 10, and 15, followed by a parameter sweep of integer values between 5 and 10. Calibration was undertaken using the root mean squared error (RMSE), where the multiplication factor which gave the lowest RMSE was chosen as the final parameter value.

**Covid-19 outcome as a function of risk factor.**   The top three comorbidities for Covid-19 are hypertension, diabetes and CVD [45, 46]. In addition, obesity and sex and are significant risk factors for poor Covid-19 outcomes [32].

The percentage of fatalities who had diabetes, CVD and were male could be compared to observed data from the UK Health Security Agency from the first wave of the pandemic [47]. The percentage of fatalities who had hypertension was compared to data from a global meta-analysis study [48]. Observed data was also available from ICNARC quantifying the percentage of Covid-19 ICU admissions by BMI [49], and a second dataset was available from Dana et al. [50]. The ICNARC dataset should be treated as the primary validation data as it includes 25,849 datapoints, compared to Dana et al. which includes data from 222 patients.

## Case study: Forecasting Covid-19 effects on a UK population

The lifelong effects of Covid-19, following various intervention strategies, were simulated on a UK population.

Five scenarios were simulated as follows:

- **Scenario 1**—Baseline: No interventions;

- **Scenario 2**—Targeted vaccinations by age: Agents aged 16 years+ are vaccinated. The vaccination programme begins at day 270 (9 months after the introduction of the virus) until the end of the simulation. Boosters are administered every 6 months until the end of the simulation.

- **Scenario 3**—Targeted vaccinations by 'clinically vulnerable' risk group: The Joint Committee on Vaccination and Immunisation (JCVI) defines clinically vulnerable people as those with a range of risk factors [51]. However, the model dataset does not include all of these risk factors, therefore here, we define clinically vulnerable agents as those with CVD, hypertension, diabetes, and morbid obesity (BMI $\geq$ 40). The vaccination programme begins at day 270 (9 months after the introduction of the virus) until the end of the simulation. Boosters are administered every 6 months until the end of the simulation.

- **Scenario 4**—Periodic lockdowns: A full lockdown is triggered once hospitalisations reach a defined threshold. This threshold was calculated by assessing the percentage of the population in England who were in hospital at the time each of the three lockdowns (26th March 2020, 5th November 2020, 6th January 2021). This gave a mean threshold of 0.027% ($\pm$0.02 sd), and a 3 month lockdown will be triggered once this number of agents are hospitalised for the duration of the simulation.

- **Scenario 5**—Periodic lockdowns with whole population vaccination programme: A full lockdown is triggered once hospitalisations reach a defined threshold (as described in scenario 4). A vaccination programme for the whole cohort begins at day 270 (9 months after the introduction of the virus) until the end of the simulation. Boosters are administered every 6 months until the end of the simulation.

## Results

### Model validation

**Hospital/ICU admissions and death rates.** The model captured the trends of the field data for hospitalisations, ICU admissions, and fatalities in terms of the outcome magnitude and the effects of the various lockdown interventions implemented by the UK government from 24th March—30th September (see Fig 3).

**Covid-19 outcome as a function of risk factor.** The modelled percentage of deaths with diabetes, hypertension, and CVD were 20.3, 45.1, and 29.1, compared to observed percentages of 21.1, 46.0, and 44.5, respectively. Whilst the modelled data for diabetes and hypertension matched the observed data within 1%, the observed data for CVD was 15.4% higher than the modelled outputs (see Fig 4a). The number of male fatalities was higher than for females both in the model and the observed data (55.5% vs 59.3%, respectively). The modelled percentage of patients in ICU with a normal, overweight, obese or morbidly obese BMI provided good predictions ($\leq$ 10% differences) of both observed data sets [49, 50] (see Fig 4b).

### Case study: Forecasting Covid-19 effects on a UK population

**Simulated effects on health-related outputs.** All vaccination scenarios resulted in the greatest reduction of hospital admissions, ICU admissions and fatalities, compared to the scenario which only simulated periodic lockdowns. The scenario with both periodic lockdowns and a whole population vaccination programme (scenario 5) was the most successful in reducing these endpoints; however, differences between scenario 2 (vaccination stratified by age), scenario 3 (vaccination stratified by clinically vulnerable individuals) and scenario 5 were marginal and all resulted in between 89 and 90% reductions in hospital and ICU admissions and fatalities compared to the baseline scenario (scenario 1). Scenario 4 which simulated periodic lockdowns with no vaccination was the least successful in reducing hospital admissions (8%

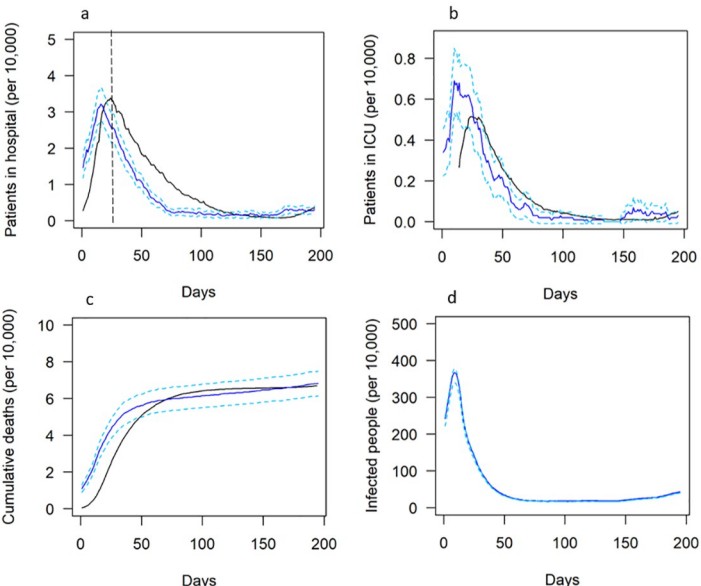

**Fig 3. Validation results.** Observed (black) and modelled (blue) data for patients in hospital (a), patients in ICU (b) and cumulative deaths (c) per 10,000 people between 20th March and 30th September 2020. Modelled data for number of infections (d) is also shown for reference. Dashed blue lines represent 95% confidence intervals and the data before the dashed black line (a) represents calibration data.

compared to baseline scenario), ICU admissions (6% compared to baseline scenario) and fatalities (8% compared to baseline scenario).

The periodic lockdown scenario (scenario 4) resulted in the greatest reduction in Long Covid cases over the lifespan of the cohort (15% compared to the baseline scenario), followed by the lockdown and vaccination scenario (scenario 5) which resulted in a 6% reduction. The remaining vaccination scenarios resulted in only marginal reductions in Long Covid cases compared to the baseline scenario (<2%).

The results are depicted in Fig 5 and Table 3.

**Simulated effects on economic-related outputs.** Compared to the baseline scenario, the periodic lockdown scenario (scenario 4) resulted in only a 7% reduction in healthcare costs. The differences between the remaining scenarios was negligible and all resulted in a 90% reduction in healthcare costs compared to the baseline scenario, with the periodic lockdown

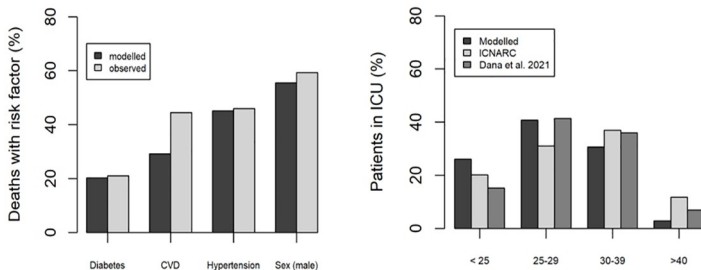

**Fig 4. Validation results.** Modelled and observed percentage of deaths by risk factor (a) and percentage of patients in ICU by BMI (b).

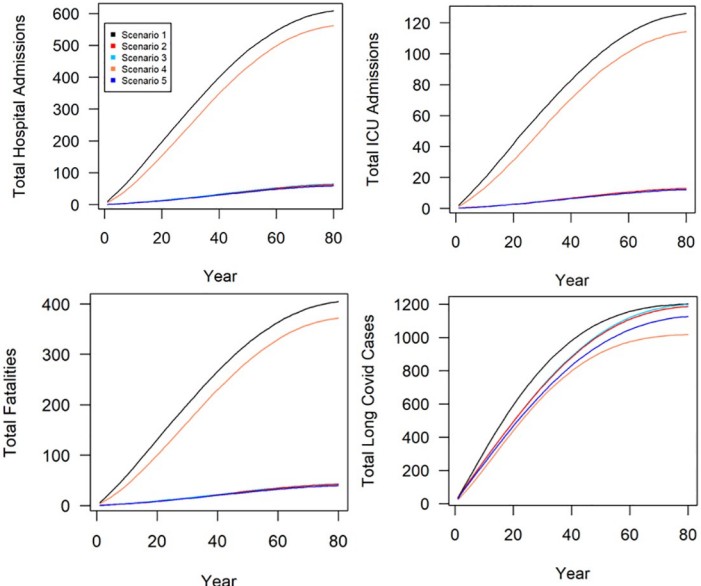

**Fig 5. Results.** Lifelong simulations of Covid-19 effects on total hospital admissions (a), total ICU admissions (b), total fatalities (c) and total Long Covid cases (d) following policy intervention scenarios. Scenario 1 is a baseline (no interventions); Scenario 2 is targeted vaccinations programme by age; Scenario 3 is targeted vaccinations by 'clinically vulnerable' risk group; Scenario 4 is periodic lockdowns; Scenario 5 is periodic lockdowns with whole population vaccination programme. Outputs represent the annual mean values of 100 simulation runs in a cohort of 1,000 agents.

and whole population vaccination programme scenario (scenario 5) resulting in the overall lowest healthcare costs (Table 2). The periodic lockdown scenario resulted in the greatest intervention costs of £173 million, followed by the periodic lockdown and whole population vaccination programme (scenario 5) which resulted in a cost of £46 million. The vaccination scenario stratified by clinically vulnerable individuals (scenario 3) resulted in the lowest intervention costs of £51,924, followed by the vaccination scenario stratified by age (scenario 2) which resulted in an intervention cost of £106,533.

Fig 6 and Table 4 show the results of the economic-related outputs.

## Discussion and conclusion

We have presented the CoronAvirus Lifelong Modelling and Simulation (CALMS) model which is an agent-based model designed to predict the lifelong health and economic effects of Covid-19, as well as to assess the impact of associated government interventions. We have demonstrated the applicability of CALMS in assessing the effects of multiple stratified

**Table 3. Mean (±sd) model outputs for healthcare related endpoints over the lifetime of the simulated cohort.**

|  | Total hospital admissions | Total ICU admissions | Total fatalities | Total Long Covid cases |
|---|---|---|---|---|
| Scenario 1 | 610.78 ±24.08 | 125.3 ±11.06 | 407.57 ±16.16 | 1,203.07 ±42.41 |
| Scenario 2 | 62.88 ±9.79 | 13.33 ±3.7 | 42.25 ±7.21 | 1,194.38 ±44.48 |
| Scenario 3 | 63.65 ±16.79 | 13.07 ±4.77 | 42.22 ±12.47 | 1,193.05 ±49.81 |
| Scenario 4 | 563.33 ± 21.66 | 117.61 ±9.75 | 375.45 ±16.27 | 1,027.81 ±47.82 |
| Scenario 5 | 58.53 ±9.26 | 11.96 ±3.70 | 39.01 ±7.32 | 1,131.86 ±85.38 |

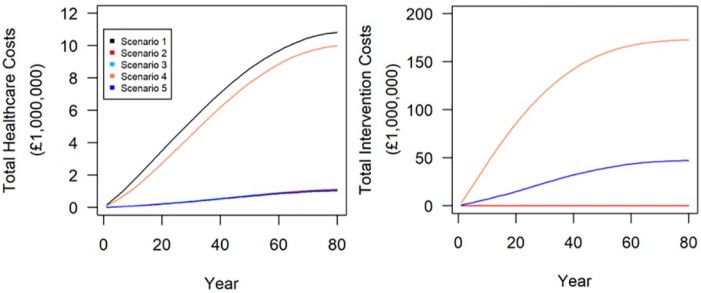

**Fig 6. Results.** Lifelong simulations of Covid-19 effects on total healthcare costs (a) and total intervention costs (b) following policy intervention scenarios. Scenario 1 is a baseline (no interventions); Scenario 2 is targeted vaccinations programme by age; Scenario 3 is targeted vaccinations by 'clinically vulnerable' risk group; Scenario 4 is periodic lockdowns; Scenario 5 is periodic lockdowns with whole population vaccination programme. Outputs represent the annual mean values of 100 simulation runs in a cohort of 1,000 agents.

vaccination scenarios and/or lockdown interventions on health- and economic-related outputs for the Covid-19 pandemic in England.

The model experimentation against UK population data showed good predictions for hospitalisations, ICU admissions and fatalities. In addition, the model provided good predictions for the number of deaths by the top three comorbidities—hypertension, diabetes and CVD [45, 46]—as well as by sex. The largest discrepancy between modelled and observed data was for predicting the percentage of deaths with CVD, where the PHE dataset reported CVD as being the greatest comorbidity. There are other global studies, however, which put CVD as a lower risk factor than hypertension and/or diabetes [52–54]. The percentage of ICU admissions by BMI also matched two observed datasets very well [49, 50]. These predictions indicated that the model is behaving correctly, that the input parameters are sufficiently accurate, and the conceptual model is suitable to provide realistic emergent outputs. We therefore determined the model was suitable for use in a forecasting case study in the UK assessing the impacts of government intervention scenarios on a population.

The model outputs from the Covid-19 case study determined that the scenario which simulated periodic lockdowns alongside a whole population vaccination programme was the most successful in reducing hospital admissions, ICU admission and fatalities. However, over the lifespan of the cohort, the differences between this scenario and the scenarios which simulated vaccination programmes without lockdowns were marginal, yet the economic cost of including periodic lockdowns was extremely high. A successful vaccination programme was key to reducing these health- related endpoints compared to the baseline scenario (by 89–90%), whilst reducing healthcare costs and having minimal intervention costs. There were only marginal differences between the vaccination programme which targeted only clinically vulnerable

**Table 4. Mean (±sd) model outputs for economic related endpoints over the lifetime of the simulated cohort.**

|  | Total healthcare costs (£) | Total intervention cost (£) |
| --- | --- | --- |
| Scenario 1 | 10,846,202 ±582,747.8 | 0 ±0 |
| Scenario 2 | 1,131,388 ±213,899.2 | 106,533.7 ±2,186.592 |
| Scenario 3 | 1,127,476 ±314,640.6 | 51,923.68 ±1,664.161 |
| Scenario 4 | 10,057,673 ±561,881 | 173,168,709 ±13,061,928 |
| Scenario 5 | 1,050,184 ±200,307.7 | 45,629,347 ±27,558,562 |

individuals, and the programme targeting all individuals aged 16+, however, the former resulted in a 51% reduced intervention cost compared to the latter. This highlights that clinically vulnerable individuals make up the vast majority of hospitalisations and fatalities, and if the supply of vaccinations is limited or there are limitations around cost, vaccinating only these individuals would be sufficient in providing population protection.

Whilst it is clear that lockdowns are necessary at the beginning of a pandemic to prevent healthcare systems becoming overwhelmed, our simulations demonstrate that they provide minimal ($\leq$15% reductions) long-term benefits for population health and are associated with high costs. The lockdown scenario, however, did have the greatest reduction in Long Covid cases compared to the baseline scenario, albeit this was only a 15% reduction. The vaccination scenarios resulted in only marginal reductions in Long Covid cases and this is because the model conservatively assumes that vaccinations do not reduce the risk of Long Covid. Research is currently unclear as to if, and to what extent, vaccinations may reduce Long Covid risk [55, 56].

Whilst there are currently a multitude of simulation studies assessing the spread of Covid-19 around the world, as well as its effects on healthcare systems [57–59], CALMS has three distinct differentiations from many of these models. Firstly, it predicts the lifelong health effects of a disease on a population. Secondly, it predicts the long-term economic costs of a disease along with the costs of interventions. Finally, it uses the QCovid algorithm to predict an individual's risk of developing severe disease outcomes based on their characteristics and comorbidities [13]. Additionally, there are now several studies assessing the cost-effectiveness of government interventions. For example, Miles et al. [60] used the same method to calculate the economic costs of a full lockdown in the UK as the current study, as well as accounting for Quality Adjusted Life Years (QALYs), and found that lockdown costs were five times higher than the benefits from avoiding the worst mortality case scenario. Whilst QALYs are not currently included in CALMS, economic outputs from the model can complement these cost-effectiveness analyses by providing insight into the long-term / lifelong economic effects of interventions.

Since Covid-19 is still a very active area of research, the model is inevitably limited by data gaps. In particular, the maximum duration of Long Covid is currently uncertain and there is no clear data describing who may be most susceptible to developing Long Covid symptoms [61]. All modelled individuals therefore currently have the same probability of developing Long Covid. There are also inherent uncertainties when it comes to predicting the long-term effects of a disease that has only been prevalent in society for a short period of time, as these long-term predictions cannot be validated. The model currently assumes that the characteristics of the virus do not change over the lifetime of the cohort. Thus, the predictions presented here represent a single variant (wild-type) of Covid-19 and demonstrate the long-term predictions of the epidemic as it stands at a given point in time. Nonetheless, a sensitivity analysis could be undertaken to explore the impact of changing key parameters, for example transmission probability or death probability. This would give an insight into how possible changes to the characteristics of the virus over time might impact on long-term health and economic outputs. In addition, the model can be easily adapted to represent other strains of Covid-19 assuming the empirical data is available.

As the coronavirus pandemic continues to evolve, more accurate data will become available and CALMS will be continually updated to reflect the most recent information. For example, QCovid is currently developed for the wild strain of Covid-19 and this would need to be considered if we aim to incorporate other variants of the virus into the model as well as different infection delay levels [62, 63]. Future considerations for the model will be to incorporate health utilities and calculate QALYs in order to provide more comprehensive economic outputs for

cost-effectiveness analysis considering cross-sectoral analysis [64]. Another potential area for future development is to assess the impacts of more long-term health related interventions, such as physical activity interventions, on Covid-19 outcomes in a population. We envisage that CALMS will be used in future pandemics, including variants of Covid-19, as a readily available tool to predict lifelong population assessments of diseases and their management.

## Supporting information

**S1 File. Data and results repository.** Data and results of the validation and case study experiments conducted with the CoronAvirus Lifelong Modelling and Simulation (CALMS) model are available at https://doi.org/10.17633/rd.brunel.19350518. The repository also includes the R code used to generate the output graphs.
(PDF)

**S2 File. CALMS code repository.** The CoronAvirus Lifelong Modelling and Simulation (CALMS) code is available on https://gitlab.com/anabrunel/calms.
(PDF)

**S3 File. CALMS ODD.** The file includes the CALMS Overview, Design concepts and Details (ODD) protocol.
(PDF)

## Author Contributions

**Conceptualization:** Kate Mintram, Anastasia Anagnostou, Nana Anokye, Edward Okine, Derek Groen, Arindam Saha, Nura Abubakar, Tasin Islam, Habiba Daroge, Yani Xue, Simon J. E. Taylor.

**Data curation:** Kate Mintram, Anastasia Anagnostou, Nana Anokye, Simon J. E. Taylor.

**Formal analysis:** Kate Mintram, Anastasia Anagnostou.

**Funding acquisition:** Anastasia Anagnostou.

**Investigation:** Kate Mintram, Anastasia Anagnostou.

**Methodology:** Kate Mintram, Anastasia Anagnostou, Simon J. E. Taylor.

**Project administration:** Anastasia Anagnostou.

**Software:** Kate Mintram, Anastasia Anagnostou.

**Supervision:** Anastasia Anagnostou.

**Validation:** Kate Mintram, Anastasia Anagnostou.

**Writing – original draft:** Kate Mintram, Anastasia Anagnostou.

**Writing – review & editing:** Kate Mintram, Anastasia Anagnostou, Nana Anokye, Edward Okine, Derek Groen, Arindam Saha, Nura Abubakar, Tasin Islam, Habiba Daroge, Maziar Ghorbani, Yani Xue, Simon J. E. Taylor.

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
