## [Decision Letter · Decision Letter 0]

9 May 2022

PONE-D-22-07757CALMS: Modelling the long-term health and economic impact of Covid-19 using agent-based simulationPLOS ONE

Dear Dr. Anastasia Anagnostou,

Thank you for submitting your manuscript to PLOS ONE. After careful consideration, we feel that it has merit but does not fully meet PLOS ONE’s publication criteria as it currently stands. Therefore, we invite you to submit a revised version of the manuscript that addresses the points raised during the review process.

This manuscript was reviewed by two reviewers and some major concerns were listed below. Please provide more data or discuss with sufficient references. Please submit your revised manuscript by Jun 23 2022 11:59PM. If you will need more time than this to complete your revisions, please reply to this message or contact the journal office at plosone@plos.org. Please include the following items when submitting your revised manuscript:A rebuttal letter that responds to each point raised by the academic editor and reviewer(s). You should upload this letter as a separate file labeled 'Response to Reviewers'.A marked-up copy of your manuscript that highlights changes made to the original version. You should upload this as a separate file labeled 'Revised Manuscript with Track Changes'.An unmarked version of your revised paper without tracked changes. You should upload this as a separate file labeled 'Manuscript'.

We look forward to receiving your revised manuscript.

Kind regards,

Wen-Wei Sung, M.D., Ph.D.

Academic Editor

PLOS ONE

Journal Requirements:

"This work was partially funded by the EU H2020 STAMINA project No. 883441."

We note that you have provided funding information. However, funding information should not appear in the Acknowledgments section or other areas of your manuscript. We will only publish funding information present in the Funding Statement section of the online submission form. 

"KM, AA, NA, EO, DG, AS, NA, TI, HD, MG, YX and ST have received funding from the EU STAMINA project No. 883441 (https://cordis.europa.eu/project/id/883441).

Reviewers' comments:

Reviewer's Responses to Questions

**Comments to the Author**

1. Is the manuscript technically sound, and do the data support the conclusions?

Reviewer #1: Yes

Reviewer #2: Yes

2. Has the statistical analysis been performed appropriately and rigorously? 

Reviewer #1: Yes

Reviewer #2: Yes

3. Have the authors made all data underlying the findings in their manuscript fully available?

Reviewer #1: Yes

Reviewer #2: Yes

4. Is the manuscript presented in an intelligible fashion and written in standard English?

Reviewer #1: Yes

Reviewer #2: Yes

5. Review Comments to the Author

Reviewer #1: This paper examines the lifelong health and economic impacts of Covid-19 and assesses the impact of related government interventions. The CoronAvirus Lifelong Modelling and Simulation (CALMS) model, an agent-based model, is used. The model is demonstrated by assessing the impact of multiple layered vaccination scenarios and/or lockdown measures on health and economic outcomes for the Covid 19 pandemic in the UK (UK or England please clarify). The results, i.e. the predictions for hospitalisations, ICU admissions and deaths, are plausible. This is a well-written paper whose strength is the use of the ABM model and the different scenarios. I have only a few comments.

Specific comments

The abstract is a bit long and only one third of the abstract deals with the results (which I find the most interesting). Report a bit more the results of the five scenarios.

“The aim of this paper is to demonstrate our approach and present the long-term health and economic impact of Covid-19 in the UK.”

Please add a sentence or two (data, method etc)-

At the end of the introduction, the authors state the aim of the paper, which is to present the long-term health and economic impact of Covid-19 in the UK using an agent-based model. Please add a sentence about the contribution of the paper to the literature. Is it the first study to use ABM in this field? Is the CALMS model new? Maybe this is the contribution (it is mentioned before):

This model takes into account the impact of individual health status on disease progression and also the long-term or lifelong health impact of Covid-19 on the population.

“CALMS input parameter names and values, sources and additional information”

How sensitive are the results when the assumptions vary?. This is a crucial issue for all ABM applications.

Check formula 1: No subscripts for i and t?

The discussion section contains all the elements, a brief summary of the results, implications, limitations and future work. There is little I could suggest for improvement.

This is a great study.

Reviewer #2: Comments on “CALMS: Modelling the long-term health and economic impact of Covid-19 using agent-based simulation” (Manuscript Number: PONE-D-22-07757)

This paper develops a Covid-19 analysis system, which consider the long-term health and economic impact. The model is interesting but still has some problems that need to be explained, which are list as following:

1. Authors should explain how to set these parameters in Tab. 2

2. This model considers the economic impact of covid-19, but it only shows the impact of the disease on cost, how would the economic impacts the cure of disease?

3. The model consider intervention. Such intervention would influence the spreading of disease, so is the transmission probability variable?

4. Please consider the following related papers

[1] Chuangxia Huang, Jie Cao, Fenghua Wen, Xiaoguang Yang. Stability analysis of SIR model with distributed delay on complex networks[J]. PLOS ONE, 2016, 11(8).

[2] Huang C, Zhang H, Cao J, et al. Stability and Hopf bifurcation of a delayed prey–predator model with disease in the predator[J]. International Journal of Bifurcation and Chaos, 2019, 29(07): 1950091.

[3] Jie Cao, Fenghua Wen, H E Stanley, Xiong Wang. Multilayer financial networks and systemic importance: Evidence from China[J]. International Review of Financial Analysis, 2021: 101882.

6. PLOS authors have the option to publish the peer review history of their article (what does this mean?). If published, this will include your full peer review and any attached files.

Reviewer #1: **Yes: **Martin Thomas Falk

Reviewer #2: No

---

## [Author Response · Author response to Decision Letter 0]

14 Jul 2022

We would like to thank the Editors and Reviewers for the opportunity to revise our article. We detailed how we address the reviewers' comments in a document attached in this submission.

---

## [Decision Letter · Decision Letter 1]

25 Jul 2022

CALMS: Modelling the long-term health and economic impact of Covid-19 using agent-based simulation

PONE-D-22-07757R1

Dear Dr. Anastasia Anagnostou,

We’re pleased to inform you that your manuscript has been judged scientifically suitable for publication and will be formally accepted for publication once it meets all outstanding technical requirements.

Kind regards,

Wen-Wei Sung, M.D., Ph.D.

Academic Editor

PLOS ONE

Reviewers' comments:

Reviewer's Responses to Questions

**Comments to the Author**

1. If the authors have adequately addressed your comments raised in a previous round of review and you feel that this manuscript is now acceptable for publication, you may indicate that here to bypass the “Comments to the Author” section, enter your conflict of interest statement in the “Confidential to Editor” section, and submit your "Accept" recommendation.

Reviewer #1: All comments have been addressed

Reviewer #2: All comments have been addressed

2. Is the manuscript technically sound, and do the data support the conclusions?

Reviewer #1: Yes

Reviewer #2: Yes

3. Has the statistical analysis been performed appropriately and rigorously? 

Reviewer #1: Yes

Reviewer #2: Yes

4. Have the authors made all data underlying the findings in their manuscript fully available?

Reviewer #1: Yes

Reviewer #2: Yes

5. Is the manuscript presented in an intelligible fashion and written in standard English?

Reviewer #1: Yes

Reviewer #2: Yes

6. Review Comments to the Author

Reviewer #1: I am satisfied with the revisions. I suggest that you mention the sensitivity analysis as future work (variations in the death rates, parameter heterogeneity etc).

Reviewer #2: The authors have revised the manuscript according to the reviewer's suggestion, and the quality of this paper is quite improved. I am happy to recommend the publication of this paper.

7. PLOS authors have the option to publish the peer review history of their article (what does this mean?). If published, this will include your full peer review and any attached files.

Reviewer #1: **Yes: **Martin Thomas Falk

Reviewer #2: No

---

## [Editor Report · Acceptance letter]

17 Aug 2022

PONE-D-22-07757R1 

CALMS: Modelling the long-term health and economic impact of Covid-19 using agent-based simulation 

Dear Dr. Anagnostou:

I'm pleased to inform you that your manuscript has been deemed suitable for publication in PLOS ONE. Congratulations! Your manuscript is now with our production department. 

Kind regards, 

on behalf of

Dr. Wen-Wei Sung 

Academic Editor

PLOS ONE